# MiR-375 and miR-21 as Potential Biomarkers of Prostate Cancer: Comparison of Matching Samples of Plasma and Exosomes

**DOI:** 10.3390/genes13122320

**Published:** 2022-12-09

**Authors:** Suzana Matijašević Joković, Zorana Dobrijević, Nevena Kotarac, Lidija Filipović, Milica Popović, Aleksandra Korać, Ivan Vuković, Dušanka Savić-Pavićević, Goran Brajušković

**Affiliations:** 1Centre for Human Molecular Genetics, Faculty of Biology, University of Belgrade, 11070 Belgrade, Serbia; 2Department for Metabolism, Institute for the Application of Nuclear Energy (INEP), University of Belgrade, 11080 Belgrade, Serbia; 3Innovative Centre, Faculty of Chemistry, University of Belgrade, 11070 Belgrade, Serbia; 4Faculty of Chemistry, University of Belgrade, 11070 Belgrade, Serbia; 5Clinic of Urology, Clinical Center of Serbia, 11000 Belgrade, Serbia

**Keywords:** prostate cancer, exosomes, plasma, microRNAs, miR-21, miR-375

## Abstract

MiR-21 and miR-375 have been reported as dysregulated in prostate cancer (PCa) in multiple previous studies. Still, variable or even opposing data for the expression of these microRNAs in PCa were found, and their potential biomarker properties remain elusive. In an attempt to clarify their significance as PCa biomarkers, as well as to compare different types of specimens as a source of relevant microRNAs, we used plasma and matching plasma-derived exosomes from patients with PCa and patients with benign prostatic hyperplasia (BPH). Plasma and exosomes were obtained from 34 patients with PCa and 34 patients with BPH, and their levels of expression of miR-21 and miR-375 were determined by RT-qPCR. We found no significant difference in the level of expression of these microRNAs in plasma and exosomes between patients with PCa and BPH. The level of exosomal miR-21 was elevated in PCa patients with high serum PSA values, as well as in patients with aggressive PCa, while for plasma samples, the results remained insignificant. For miR-375, we did not find an association with the values of standard prognostic parameters of PCa, nor with cancer aggressiveness. Therefore, our results support the potential prognostic role of exosomal miR-21 expression levels in PCa.

## 1. Introduction

The most recent update on cancer statistics from the Global Cancer Observatory (GCO) in the GLOBOCAN online database identifies prostate cancer (PCa) as the third most commonly diagnosed cancer worldwide in 2020. Furthermore, PCa was reported as the most frequently diagnosed cancer among males in 12 out of 20 world regions [1]. According to these statistics and the projected costs associated with PCa treatment, this disease represents one of the major global health burdens, as well as one of the major socio-economic burdens related to human health [2]. The global trend of the increasing incidence of PCa and its metastatic type, as well as the difficulties in the differentiation of indolent and localized versus highly aggressive types of tumors, makes PCa management both challenging and expensive. Therefore, novel, validated biomarkers for localized PCa are needed to improve risk stratification and to aid in the diagnostics and decision making, by enabling the more precise monitoring of the biological behavior of the tumor [3].

Among the most recognized candidates for PCa monitoring through liquid biopsy are microRNA molecules from cell-free body fluids and plasma and serum-derived extracellular vesicles (EVs) [4,5,6,7]. The main characteristics that qualify microRNAs as potential PCa biomarkers are their chemical stability, the minimal invasiveness of sample acquisition, and their regulatory properties and changes in expression related to malignant transformation [6]. Still, the results obtained to date are inconclusive and dependent on the pre-analytical factors and analytical approaches, as well as on the selection of sample type. The lack of reproducibility refers not only to opposing results related to single microRNA candidates and to differences between the panels of microRNA defined as potential biomarkers of PCa in different studies, but also to the direction of changes in the expression of certain microRNA molecules [7].

The results obtained by RNA-seq and microRNA arrays identified only a small number of microRNAs with partially reproducible up- or downregulation in plasma of PCa patients. MiR-21 and miR-375 are among the rare microRNAs reported as dysregulated microRNAs in PCa in multiple studies. Opposing data for the expression of these microRNAs in PCa have been reported, even for the same types of specimens [7]. For instance, even for cancerous tissue, the reports on the direction of dysregulation of miR-21 were contradictory [7]. Therefore, the diagnostic utility of free and EV-incorporated miR-21 and miR-375 in PCa remains inconclusive. Even more importantly, their potential significance for disease monitoring and usefulness as prognostic biomarkers of PCa was evaluated in a relatively small number of studies with diverse, or even conflicting, findings [4]. Based on previous findings, these two microRNAs were selected as the most plausible candidates, and our selection matched the prevailing conclusions of articles summarizing the data from original studies focusing on the role of circulating microRNAs as PCa biomarkers [4,5,8]. The selection criteria were not based on their functional significance, even though both of these microRNAs are found to contribute to PCa progression [8]. The involvement of mir-21 in the evasion of apoptosis and the proliferation rate control in PCa were reported, in addition to its role in epithelial–mesenchymal transition, cancer invasion, and PCa metastasis [8]. Furthermore, this microRNA is involved in androgen signalization through a positive feedback regulatory mechanism [9]. Mir-375 was also implicated in tumor invasion and PCa metastasis, while a dual role of this microRNA in PCa was suggested, dependent on cellular context and conditioned by androgen regulation [7,10].

In an attempt to clarify the diagnostic significance of miR-21 and miR-375 in PCa, we used two different types of matching samples obtained from Serbian PCa patients and controls diagnosed with benign prostatic hyperplasia (BPH). Apart from plasma samples, EVs extracted from plasma were used as a source of microRNAs, and the correlation in the expression levels of miR-21 and miR-375 between these two types of specimens were assessed. Furthermore, one of the main aims of the present study was to evaluate the association of the expression levels of miR-21 and miR-375 with the values of standard prognostic parameters of PCa, as well as with the risk for cancer progression.

## 2. Materials and Methods

### 2.1. Patients: Clinical Data and Sample Collection

This study included 35 patients with PCa and 34 patients with BPH collected during the period 2020–2021 at the Urology Clinic of the Clinical Center of Serbia. Research was approved by the ethics committee of the involved medical center with the permit obtained in July 2020 (code 570/8). Written information consent was obtained from all study participants, and the research was conducted in accordance with the Declaration of Helsinki.

The diagnoses of study participants were established using the standard clinical diagnostic procedure, which included the measurement of the serum level of prostate-specific antigen (PCA), digital rectal examination (DRE), transrectal ultrasound scan (TRUS), and prostate biopsy, followed by bone radiography. The Gleason score and TNM classification system was used to define histopathological type and clinical stage of cancer, respectively. Patients’ age and clinical data were obtained from medical records, upon obtaining explicit freely given informed consent from each participant.

Patients with PCa were selected into groups based on the values of standard prognostic parameters: initial serum PSA (PSA ≤ 20 ng/mL; PSA > 20 ng/mL), Gleason score (GS ≤ 6; GS ≥ 7), and clinical stage (T1 and T2; T3 and T4). According to recommendations by the European Association of Urology (EAU), patients with PCa were selected into the following groups based on the values of standard prognostic parameters of disease progression: less aggressive group (low- to intermediate-risk group according to EAU criteria: PCa patients with PSA ≤ 20 ng/mL, GS ≤ 7, and clinical stage T1 and T2) and aggressive group (high-risk group according to EAU criteria: PCa patients with PSA > 20 ng/mL, GS > 7, or clinical stage T3 and T4). Patients with the presence of distant metastasis were sorted into the aggressive group. Classification of PCa patients according to clinical data is presented in Table 1. In addition, the PCa patients with the missing data on prognostic parameters were excluded from classification.

Fresh blood samples were collected into commercially available EDTA-treated tubes and subjected to the conventional method for separating the cellular fraction from plasma within 30 min of blood draw. Plasma samples were obtained by centrifugation of peripheral blood samples for 15 min at 2000× *g* using a refrigerated centrifuge, followed by an immediate transfer of the supernatant into a clean cryotube and storage of plasma at −80 °C.

### 2.2. Plasma Exosome Isolation

Exosomes were isolated by the newly described original method based on immune-affinity chromatography using a nanobodies (single-domain antibodies, VHHs)-coated matrix that recognizes exosome epitopes in their native conformation. The protocol was previously published in Filipović et al. and described in detail [11]. The relative concentration of exosomes was determined through the measurement of protein amount, employing the Qubit Fluorometer and Qubit Protein Assay Kit (Thermo Fisher Scientific, Waltham, MA, USA). The presence of exosomes in the sample isolate was confirmed by flow cytometry and transmission electron microscopy.

### 2.3. Flow Cytometry

Bead-based flow cytometry was used for characterizing exosomes by coating beads with the extracted extracellular vesicles in PBS (overnight at 4 °C, 40 µg of total protein). After washing with PBS (three times) and blocking (30 min at RT with 200 mM glycine, 30 min in PBS plus 5% (*w*/*v*) skimmed milk), beads were washed again with PBS and incubated with antibodies against exosome marker for 1 h at 37 °C with anti-CD9 antibodies (dilution 1:5). Beads were analyzed using a BD FACSCalibur (BD Biosciences), collecting around 1000 events/s, and a blue solid-state 200 mW laser at 488 nm was used for excitation. The emission corresponding to labelled antibodies was detected at 561 nm (FL2, PE, for anti-human CD9). Latex beads without the exosome coating blocked with glycine and milk were used as a control.

### 2.4. Transmission Electron Microscopy

Transmission electron microscopy (TEM) was used to characterize the extracted vesicles after sample negative staining. After the adsorption of the exosome sample to a carbon-coated grid, fixation was performed by using 2.5% glutaraldehyde. Exosomes were contrasted by 1% phosphotungstic acid and air-dried. Electron micrographs were collected using a Philips CM12 electron microscope (Philips, Eindhoven, The Netherlands) equipped with the digital camera SIS MegaView III (Olympus Soft Imaging Solutions, Münster, Germany).

### 2.5. RNA Extraction

Total RNA, including microRNA, was extracted from the exosome isolate and plasma using miRNeasy serum/plasma kit (Qiagen, Hilden, Germany), according to manufacturer’s instructions. The synthetic spike-in control cel-miR-39 was introduced at the beginning of the isolation procedure, after the addition of QIAzol Lysis Reagent, and it was used as a normalizer in the downstream analysis. RNA was eluted with 15 µL of RNase-free water. NanoDrop Spectrophotometer (Thermo Fisher Scientific, Waltham, MA, USA) was used to assess the quality and quantity of the extracted RNA. Isolates of RNA were stored at −80 °C until further usage.

### 2.6. Reverse Transcription and RT-qPCR Analysis

An amount of 1.8 μL of each RNA sample was converted to cDNA using TaqMan™ Advanced miRNA cDNA Synthesis Kit (Applied Biosystems™, Waltham, MA, USA). After the reverse transcription, cDNA was pre-amplified using Universal miR-Amp Primers from the same kit, in order to improve the detection of low-expressing microRNA targets. The expression analysis of two microRNAs (hsa-miR-375-3p, hsa-miR-21-5p) was performed using TaqMan™ Advanced miRNA Assays (assay IDs: 478074_mir and 477975_mir, Applied Biosystems™, Waltham, MA, USA), according to manufacturer’s protocol. qPCR quantification was carried out by StepOne™ Real-Time PCR System (Applied Biosystems™, Waltham, MA, USA) with a temperature profile that consisted of the enzyme activation at 95 °C for 20 s, 40 cycles that included denaturation at 95 °C for 1 s, and a primer annealing/elongation step at 60 °C for 20 s.

The delta-delta Ct method was used for the calculation of the relative expression of microRNAs. To calculate the relative fold gene expression, 2^−∆∆Ct^ values were compared between PCa and BPH patients. Relative microRNA expression levels were normalized against the combination of spike-in control cel-miR-39 and the endogenous normalizer hsa-miR-1228 (assay IDs: 478643_mir and 478293_mir, Applied Biosystems™). The mean Ct value of the reference genes was subtracted from the Ct value of the target microRNAs to calculate ΔCt. ΔΔCt values were calculated by subtracting the average ΔCt value of the BPH group from ΔCt values corresponding to each sample.

### 2.7. Statistical Analysis

The statistical analyses were performed with statistical software RStudio 2021.09.1. The Shapiro–Wilk test was employed for assessing the normality of the distribution of results, while the F-test was used to analyze the equality of variances in study groups. Two-tailed Student’s *T*-test was used for paired comparisons of normally distributed results, while the Mann–Whitney U test was applied to analyze the data that significantly deviated from the normal distribution. *P* values < 0.05 were considered as statistically significant. The results of microRNA expression quantification were correlated with serum PSA levels using a linear regression analysis. The results were presented as Pearson correlation coefficients (r) and the corresponding *p* values. The area under the curve (AUC) was calculated through a receiver-operating characteristic (ROC) curve analysis to determine the sensitivity and specificity of the potential biomarkers. Patients with missing data on some of the prognostic parameters were excluded from the association analyses relevant for that exact parameter.

## 3. Results

Classification of PCa patients according to the available clinical and pathological data is presented in Table 1. The majority of PCa patients had an initial serum PSA value lower than 20 ng/mL (71.4%) and prostate cancer with a Gleason score of 7 (62.8%). Distant metastases were detected in only 8.6% of participants included in the study. According to the proposed classification criteria, the majority of the patients were diagnosed with aggressive PCa (51.4%).

To confirm the yield and the quality of exosomes, the plasma EV samples from 5 PCa and 5 BPH patients were characterized by morphological and protein structure analyses. TEM analysis confirmed the majority of observed vesicles in the analyzed samples were round and cup-shaped, with sizes between 30 and 100 nm that correspond to the shape and the size of exosomes (Figure 1a). Flow cytometric analysis showed that the vesicle isolates from both sample types (PCa or BPH) were positive for the distinguished exosomal marker, CD9 (Figure 1b). No significant difference was found between PCa and BPH in exosome counts based on the amount of CD9 signals (16.03 ± 0.019%; 18.22 ± 0.025%, *p* = 0.1).

The real-time PCR quantification results suggested no significant difference in miR-21 and miR-375 expression between PCa and BPH patients for both plasma samples and plasma-derived exosomes (Figure 2). Correlation tests showed the lack of statistical significance when comparing the relative expression of miR-375 in matched plasma and exosome samples (r = −0.045 and *p* = 0.722, for all participants; Figure 3a, Table 2). Even though correlation between the expression levels of miR-21 in the whole plasma and the plasma-derived exosomes was statistically significant, the obtained Pearson’s correlation coefficient of 0.28 suggests only a low level of positive correlation (*p* = 0.025, all participants; Figure 3b, Table 2). However, when correlation analyses were restricted to a specific diagnosis, the significant r value was found only for BPH patients and was lost in PCa (r = 0.386 and *p* = 0.032 for miR-21, Table 2).

When patients with PCa were segregated according to the values of standard prognostic parameters and the risk of cancer progression, no significant differences were found between relevant PCa subgroups in the level of expression of both analyzed microRNAs in whole plasma (Table 3). Similarly, exosomal miR-375 levels did not differ between subgroups of PCa patients formed according to the values of standard prognostic parameters and PCa aggressiveness (Table 3). On the other hand, when classification of PCa patients was made according to the initial serum PSA values, the obtained results revealed that the expression levels of exosomal miR-21 were significantly higher in the PCa patients with a PSA score < 20 ng/mg compared with PCa patients with a PSA score > 20 ng/mL (*p* = 0.012, Table 3, Figure 4). By transforming the dCt values into fold change, the average expression level of miR-21 was in average 6.09-fold in the group of PCa patients with a PSA score < 20 ng/mg, compared with the expression level of this microRNA in the PCa patients with a PSA score > 20 ng/mL.

Furthermore, the comparison of exosomal miR-21 expression levels between PCa patients with less aggressive and aggressive PCa demonstrated the significant difference (*p* = 0.028, Figure 5a). The average level of miR-21 was 2.7-fold lower in the aggressive group compared to the group of PCa patients with less aggressive disease. The AUC value for the ROS curve designed for assessing the potential of exosomal miR-21 to predict the aggressive PCa was 0.74, with the corresponding *p* value of 0.017.

## 4. Discussion

The main issues in the modern medical practice related to PCa management have directed the focus of a substantial number of investigators, as well as significant research efforts and funding, towards the discovery of novel PCa biomarkers. The major tendencies of such research are to define minimally invasive specific biomarkers which could aid in distinguishing benign from malignant prostatic tumors and, more importantly, improve the distinction of less aggressive and aggressive localized PCa and assist in the process of active monitoring PCa patients [6].

Due to their diverse and versatile roles in the regulation of major pathways and processes involved in the maintenance of cellular homeostasis, microRNAs emerged as one of the important actors in the molecular pathogenesis of PCa. Accordingly, these non-coding RNA molecules and genes involved in microRNA-related regulatory mechanisms were also recognized as potential sources of novel PCa biomarkers [7]. Nevertheless, the findings of molecular genetic studies focusing on microRNA-related genes, as well as of studies analyzing the biomarker properties of circulatory microRNA in PCa, remain inconclusive. Since our previous case–control studies on microRNA-related genetic variants and PCa provided promising results [12,13,14,15,16], we further expanded our research topic by analyzing the expression of selected candidate microRNAs in plasma and EVs extracted from whole plasma samples obtained from Serbian patients diagnosed with PCa.

The matched samples of whole plasma and exosomes isolated from plasma were used as sources of circulatory microRNAs in order to determine which type of specimens would be a more appropriate representative of the tumor phenotype. The rationale for the selection of plasma exosomes as carriers of potential biomarker microRNAs was based on the findings suggesting that malignant cells produce a large quantity of EVs, which circulate in body fluids and whose cargo could resemble the molecular composition of parent cells [17]. Regulation of loading mechanisms further enables the selective preferential inclusion of molecular constituents relevant to tumor progression, supporting the proposed significance of exosomes as putative surrogates of tumor cells in relation to their properties as potent mediators of intercellular communication [17,18]. Such an approach was previously used in a single study on PCa, which has a significant limitation related to the exclusion of PCa patients with a GS of 7, who represent the vast majority of the PCa group, thereby potentially influencing the study results [19]. With our study design, we tended to overcome such limitation and to test our initial hypothesis about the potentially better performance of exosomes vs. plasma as a source of PCa-relevant microRNAs. Since our goal was to examine the potential of these microRNAs in the discrimination between benign and malignant prostate tumors, which is a major problem in PCa-related clinical practice, BPH patients were used as a control group, instead of self-reported healthy controls who did not undergo the standard diagnostic procedure for PCa or had a negative PSA testing result. Furthermore, our study aimed to assess the potential prognostic significance of miR-21 and miR-375, which was far less frequently analyzed than their diagnostic potential.

In our present study, we analyzed miR-21 and miR-375 as potential biomarkers useful for the discrimination of PCa and BPH by comparing their expression levels in the cases and controls. Previously, miR-21 was found to be androgen-regulated, and its role in the promotion of PCa growth, epithelial–mesenchymal transformation in prostate tissue, evasion of apoptosis, and cancer invasiveness was established [9,20,21,22,23,24,25]. Furthermore, this microRNA was significantly upregulated in urinary exosomes or total urine samples of PCa patients, compared to healthy men [26,27,28]. Similarly, miR-375 was upregulated in PCa tissue, which stimulated cancer growth [29,30]. Another study, which analyzed the function of miR-375 in PCa cell lines, suggested the dual role of this microRNA in the carcinogenesis of PCa, depending on the cellular context [10]. When it comes to miR-21 and miR-375 in serum and plasma of PCa patients, as well as in circulatory EVs, the results obtained to date are discordant [7]. Our findings did not support the differential expression of these microRNAs in whole plasma samples and EVs from plasma between PCa patients and controls. Since we used BPH patients as a control group, differences between our results and some of the previous results could be explained by the choice of the controls. Furthermore, the results are dependent on the recruitment methodology and the selection of PCa patients, since the composition of this group, in terms of disease stage and aggressiveness, could influence the findings. Still, our results differ from the findings of Endzeliņš et al., who used a somewhat similar approach in assessing the significance of miR-21 and miR-375 as PCa biomarkers [19]. In their study, miR-375 from whole plasma and miR-21 from plasma-derived EVs showed diagnostic biomarker potential [19]. However, their PCa group did not include patients with a GS of 7, and a different methodological approach was used for EV extraction and microRNA quantification, including a different normalization strategy [19]. Our findings suggesting the lack of or a low-level correlation between the expression of microRNAs in whole plasma and EVs is consistent with the results of a previous study, which also stated that microRNA profiles of these two types of samples significantly differ [19]. One of the possible explanations for the differences in the correlation of plasma/EV levels of miR-21 between PCa and BPH groups could be that the low level of correlation was lost in PCa due to the higher representativeness of PCa by exosomal content and by the dependence of exosomal miR-21 incorporation on PCa stage and its metastatic potential. Therefore, the correlation observed in BPH, although relatively low-level, could be a result of a steady-state expression of plasmatic and exosomal miR-21 in benign tumor, which is potentially greatly affected by the predominant exosomal incorporation of upregulated miR-21 in aggressive malignant prostatic cells.

In order to test the supposed association of circulatory miR-21 and miR-375 levels with PCa stage and/or aggressiveness, we compared the relative expression of these microRNAs between patient groups formed according to the values of the standard prognostic parameters of PCa. Previously, the miR-21 level in prostate tissue was recognized as an independent predictor of biochemical recurrence of PCa and was associated with clinical and pathological characteristics of cancer [31,32,33]. The plasma level of miR-21, or microRNA panels which include this microRNA, was reported to be associated with PCa aggressiveness and clinicopathological characteristics but with variable results related to specific grading/staging criteria and the parameter used [34,35,36]. Our results regarding miR-21 in plasma samples are, therefore, to some extent, discordant with some of the previous findings. Still, one should bear in mind that different grading/staging and risk assessment criteria were used and that insignificant results regarding the difference in the expression levels in patients with various GSs are matched with previously published results [19,35]. The results presented in the current study, suggesting that the miR-21 expression level in plasma-derived exosomes is higher in patients with high initial serum PSA values, as well as in PCa patients with aggressive cancer, are in line with the supposed oncogenic properties of this microRNA, as well as with the proposed better PCa-representative properties of exosomes, compared to whole plasma. Taking into account the observed AUC value for the ROC curve constructed for cancer aggressiveness, miR-21 qualifies as a potential biomarker of aggressive PCa, which should be validated in a larger patient group, in addition to other EV-microRNAs as part of a panel, and in combination with standard prognostic parameters of PCa.

Even though miR-375 in plasma/serum and EVs was more commonly reported as a potential PCa biomarker compared to miR-21, not many studies suggested the association of this microRNA with disease prognosis, stage and aggressiveness. These studies were mostly dealing with castration-resistant PCa and recurrent disease and reported miR-375 as a predictor of survival or disease recurrence [37,38,39]. Other findings of an association between miR-375 levels and clinical characteristics of PCa are rarely reported and not matched among studies [35,39,40,41,42]. Our results, which did not suggest the difference in miR-375 levels between PCa patients stratified according to standard prognostic parameters are, therefore, partially consistent with previous findings, including the ones obtained in the earlier-mentioned study of Endzeliņš et al. [19].

The main limitation of the present study is a relatively small number of participants, as well as a small pool of analyzed microRNAs. Therefore, the findings need to be interpreted with caution, and the results need to be evaluated in a larger study group, while inclusion of other microRNA candidates could result in a panel with higher diagnostic or prognostic properties. Still, our findings support the role of exosomal miR-21 as a predictive factor for the assessment of PCa aggressiveness. Our results also illustrate the potential of exosomes to outperform the matching total plasma samples as a source of PCa-related microRNAs with biomarker properties. Therefore, our future research perspectives are related to defining a panel of exosomal surface antigens which would enable the enrichment and a more precise characterization of PCa-derived EVs and their cargo. The prospective study design and an adequate follow-up with multiple sampling time-points would also enable the assessment of changes of candidate microRNA levels in relation to disease progression and therapy, as well as the evaluation of fluctuations related to potential confounders.

## 5. Conclusions

The present study demonstrated that plasma exosomal miR-21 may serve as a more accurate noninvasive prognostic biomarker compared with the whole plasma miR-21 for active monitoring of PCa patients. Exosomal miR-21 may be used as predictive factor for the detection of aggressive PCa.

## Figures and Tables

**Figure 1 genes-13-02320-f001:**
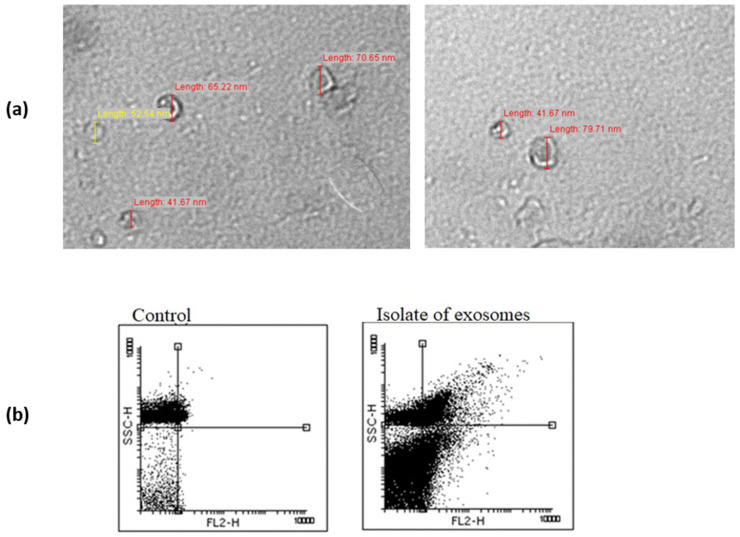
(**a**) Transmission electron microscopy (TEM) of exosomes. The observed appearance of EVs corresponds to the morphology and size range of exosomes. (**b**) Flow cytometry analysis detected the expression of the surface protein marker of exosomes, CD9.

**Figure 2 genes-13-02320-f002:**
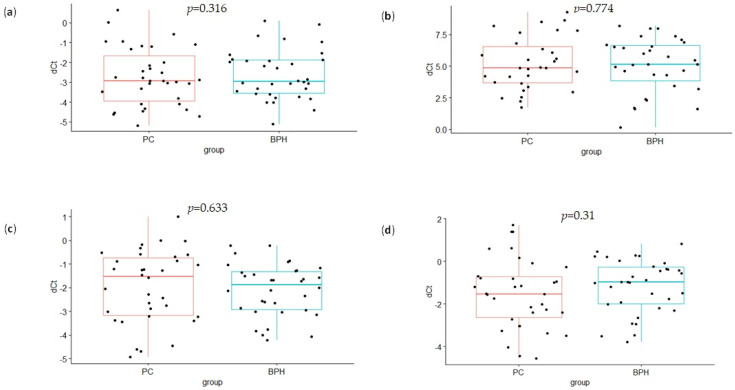
Differences in the expression levels of exosomal and whole-plasma microRNAs between PCa and BPH patients. Quantitative real-time polymerase chain reaction was executed to analyze the relative expression of miR-375 and miR-21 in exosomes (**a**,**b**, respectively) and in plasma (**c**,**d**, respectively). Reactions were normalized against the spike-in and endogenous controls (dCt, *y*-axis). Data are shown as interquartile ranges with means ± SD. A Student’s *t*-test was used to determine the statistical significance. PC—prostate cancer; BPH—benign prostatic hyperplasia; dCt—delta cycle threshold.

**Figure 3 genes-13-02320-f003:**
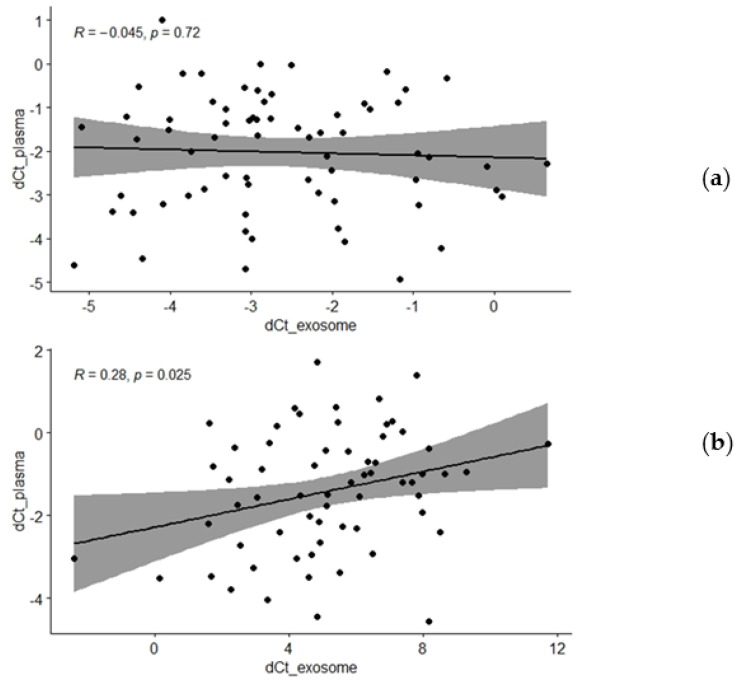
Correlation of miR-375 and miR-21 (**a**,**b**, respectively) expression levels between whole plasma and plasma-derived exosomes of all participants (joint PCa and BPH groups). Relative expression of each microRNA was normalized against the spike-in and endogenous controls in plasma and exosome samples (dCt_plasma, *y*-axis; dCt_exosomes, *x*-axis). Pearson correlation coefficients (R) and the matching *p* values corresponding to the linear regression analysis are shown on scatter plots. dCt—delta cycle threshold.

**Figure 4 genes-13-02320-f004:**
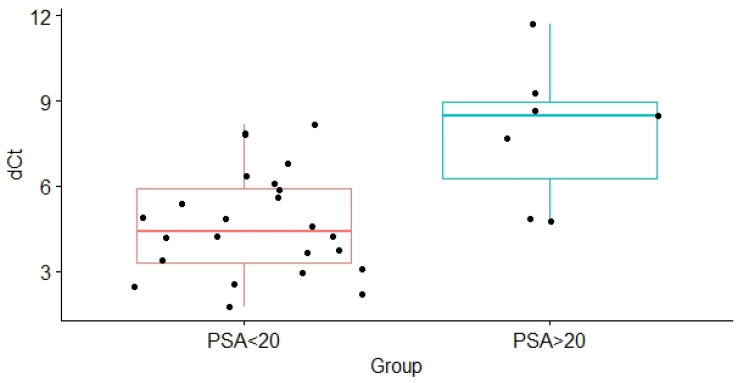
Difference in the expression levels of exosomal miR-21 between PCa patients with PSA values ≤ 20 and PCa patients with PSA values > 20. Relative miR-21 expression in subgroups was normalized against the spike-in and endogenous controls (dCt, *y*-axis). Data are shown as interquartile ranges with means ± SD. A Student’s *t*-test was used to determine statistical significance. dCt—delta cycle threshold; PSA—prostate specific antigen.

**Figure 5 genes-13-02320-f005:**
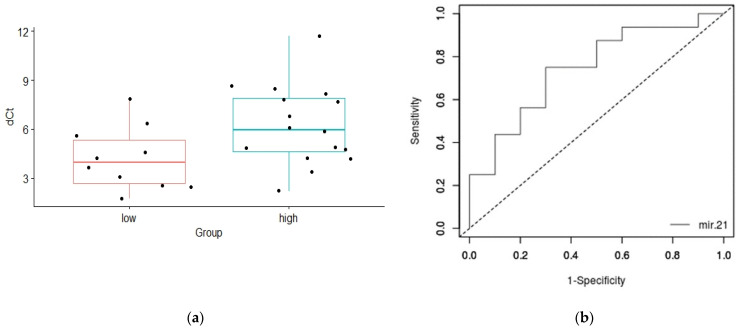
Exosomal mir-21 expression and PCa aggressiveness: (**a**) Difference in the expression levels of exosomal miR-21 between PCa patients with less aggressive and aggressive PCa. Relative miR-21 expression in subgroups was normalized against the spike-in and endogenous controls (dCt, *y*-axis). Data are shown as interquartile ranges with means ± SD. A Student’s *t*-test was used to determine statistical significance; (**b**) ROC analysis of exosomal miR-21 was performed to differentiate among PCa patients with less aggressive and aggressive cancer. The corresponding AUC value was 0.74, with *p* of 0.017. dCt—delta cycle threshold.

**Table 1 genes-13-02320-t001:** Classification of PCa patients based on the values of standard prognostic parameters and on the risk of disease progression.

Standard Prognostic Parameter	PCa Patients—*n* (%)
PSA at diagnosis	
<10 ng/mL	15 (42.8)
10–20 ng/mL	10 (28.6)
>20 ng/mL	8 (22.8)
Biopsy Gleason score	
<7	7 (20)
=7	22 (62.8)
>7	4 (11.4)
TNM stage	
T1/T2	13 (37.1)
T3/T4	16 (45.7)
Metastases	
Distant (M+)	3 (8.6)
Regional (N+) or not detected	26 (74.3)
Risk of progression	
Less aggressive	10 (28.6)
Aggressive	18 (51.4)

Abbreviations: PSA—prostate-specific antigen.

**Table 2 genes-13-02320-t002:** Correlation of microRNA expression levels between whole plasma and plasma-derived exosomes.

Groups of Participants	*r* Value	*p* Value
miR-375		
All participants	−0.045	0.722
BPH	−0.341	0.049
PCa	0.139	0.438
miR-21		
All participants	0.281	0.025
BPH	0.386	0.032
PCa	0.228	0.209

Abbreviations: PCa—prostate cancer; BPH—benign prostatic hyperplasia.

**Table 3 genes-13-02320-t003:** Correlation of microRNA expression levels between whole plasma and plasma-derived exosomes.

Groups of Participants	dCt Mean ± SD	*p* Value
exosomal miR-375		
PSA ≤ 20/PSA > 20	−2.57 ± 2.58/−3.02 ± 0.99	0.453
T1T2 stage/T3T4 stage	−2.68 ± 1.59/−3.27 ± 1.03	0.271
GS < 7/GS ≥ 7	−2.36 ± 3.28/−2.89 ± 1.84	0.395
less aggressive/aggressive	−2.95 ± 1.58/−2.66 ± 3.08	0.615
exosomal miR-21		
PSA ≤ 20/PSA > 20	4.69 ± 1.84/7.92 ± 2.47	0.012
T1T2 stage/T3T4 stage	4.67 ± 2.15/6.08 ± 2.43	0.124
GS < 7/GS ≥ 7	5.26 ± 2.03/5.07 ± 2.43	0.852
less aggressive/aggressive	4.21 ± 1.93/6.23 ± 2.43	0.028
plasma miR-375		
PSA ≤ 20/PSA > 20	−3.15 ± 2.31/−2.88 ± 0.34	0.486
T1T2 stage/T3T4 stage	−2.22 ± 1.45/1.82 ± 1.48	0.476
GS < 7/GS ≥ 7	−1.95 ± 1.56/−1.87 ± 1.44	0.905
less aggressive/aggressive	−1.84 ± 1.96/−2.59 ± 1.46	0.172
plasma miR-21		
PSA ≤ 20/PSA > 20	−1.34 ± 3.05/−1.89 ± 1.89	0.426
T1T2 stage/T3T4 stage	−2.04 ± 1.13/−1.49 ± 1.74	0.333
GS < 7/GS ≥ 7	−1.68 ± 1.49/−0.89 ± 2.13	0.926
less aggressive/aggressive	−1.64 ± 2.84/−1.77 ± 1.30	0.836

Abbreviations: PCa—prostate cancer; BPH—benign prostatic hyperplasia; PSA—prostate specific antigen; GS—Gleason score.

## Data Availability

The data supporting the findings of this study are available from the corresponding author, upon reasonable request.

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
