# Peer review of "MiR-375 and miR-21 as Potential Biomarkers of Prostate Cancer: Comparison of Matching Samples of Plasma and Exosomes"

_genes, 2022, doi:10.3390/genes13122320_

Round 1

Reviewer 1 Report

The study "Mir-375 and miR-21 as potential biomarkers of prostate cancer: comparison of matching samples of plasma and exosomes" by Matijašević et al. is an interesting study where they have explored the level and expression of miR-21 and miR-375. I do not recommend this study to publish in genes because of the following reasons.

1- What is the logic of two randomly selected miRNAs detection in PCs? Are they related? 

2- The sample size is extremely low especially when there are sub-groups.

3- Healthy controls were not used in the study so these results cannot be trusted. 

4- Biological role of these miRNAs is not discussed in the manuscript.

5- Almost all results are non-significant except for one parameter. This could be the reason for the low number of samples used in the study. 

Reviewer 2 Report

Brief Summary: The aim of the study by Matijasevic et al., was to reevaluate the prognostic potential of the microRNAs (miRNAs), miR-21 and miR-375 for prostate cancer risk and progression. The authors assessed the expression of these miRNAs in whole plasma and plasma-derived exosomes of patients with prostate cancer or with prostatic benign neoplasia (BPH, a pre-malignant form of the disease). They show no significant differences between the expression levels of either mRNAs between prostate cancer and BPH patients, as well as no significant association between miR-375 expression and disease risk or progression. The only statistically significant result of this study was that a positive association between exosomal miR-21 expression and higher risk for prostate cancer, which partly confirms data from previous published studies. The manuscript would benefit from clarifications and additional analyses to be considered for publication.

Strengths of the study: 

·       Analyses to assess the differences in prognostic value of whole serum and exosomal miRNAs for prostate cancer patients

·       Confirmatory prognostic role for serum-derived miR-21 expression in prostate cancer

Weaknesses of the study:

·       Confirmatory studies that lack novelty

·       Analyses are unclear and somewhat redundant in some points

Comments:

·       The statement that miRNAs are ‘amongst the most recognized candidates for prostate cancer monitoring” is not well supported. The authors should cite relevant literature that supports this claim. [major]

·       The aim of their study was to “clarify” the inconclusive previous findings on the role of miRNAs in prostate cancer risk and prognosis but did not provide any novel information. [minor]

·       It is unclear how the present study differs from previously published ones (cited in the manuscript), hence the “confirmatory” label. The authors may want to clarify the differences/strengths of their study in comparison to others, rather than just comparing the results. [major]

·       The primer sequences for the TaqMan probes should be included in the methods. [minor]

·       In Table 1, there is a discrepancy in the number of patients included for the analyses of each clinical parameter. For example, the PSA levels and Gleason score for 33 patients is shown but for TNM/Metastasis stagging only 29 patients are included and for Risk assessment only 28. The authors should clarify this since it is unclear how many and which patients were used for the subsequent analyses. [major]

·       The data for comparison between exosome number quantification in prostate cancer and BPH patients, albeit not significant, is not shown. [major]

·       The authors mention that correlation analyses in Table 2, were significant only for the BPH patients but they never discuss the implications of these results in the discussion. [major]

·       Labeling in Figure 3 is unclear. Which patients were analyzed here? Also, Figure 3 and Table 2 appear to provide similar information and are redundant. [minor]

Reviewer 3 Report

It's a significant work, conpliments to he authors.

A single question?...The level of expression of these microRNA in plasma in PCa patients are influenced by medical treatment?...Thanks.

Round 2

Reviewer 1 Report

Dear Authors,

Thank you for incorporating all suggested comments.

Author Response

We would like to thank the reviewer for the comments and suggestion.

Reviewer 2 Report

Overall, the authors address the points raised here and clarify some discrepancies, especially on the novelty of the results. However, one of their replies was inadequate and require further clarifications:

·       The authors do point out in the abstract that “The level of exosomal miR-21 was elevated in PCa patients with high serum PSA values, as well as in patients with high risk PCa, while for plasma samples the results remained insignificant”, suggesting that indeed, higher miR-21 expression associates with high(er) risk PCa. If the authors do not intend for this conclusion to be withdrawn, they should rephrase throughout the manuscript. [minor]

Author Response

We would like to thank the reviewer for the comments and suggestion. We have rephrased based on the suggestion.